# Ubiquitin transfer by a RING E3 ligase occurs from a closed E2~ubiquitin conformation

Emma Branigan 🄳 [1], J. Carlos Penedo 🄳 [2,3✉] & Ronald T. Hay 🄳 [1✉]

Based on extensive structural analysis it was proposed that RING E3 ligases prime the E2~ubiquitin conjugate (E2~Ub) for catalysis by locking it into a closed conformation, where ubiquitin is folded back onto the E2 exposing the restrained thioester bond to attack by substrate nucleophile. However the proposal that the RING dependent closed conformation of E2~Ub represents the active form that mediates ubiquitin transfer has yet to be experimentally tested. To test this hypothesis we use single molecule Förster Resonance Energy Transfer (smFRET) to measure the conformation of a FRET labelled E2~Ub conjugate, which distinguishes between closed and alternative conformations. We describe a real-time FRET assay with a thioester linked E2~Ub conjugate to monitor single ubiquitination events and demonstrate that ubiquitin is transferred to substrate from the closed conformation. These findings are likely to be relevant to all RING E3 catalysed reactions ligating ubiquitin and other ubiquitin-like proteins (Ubls) to substrates.

---

[1] Centre for Gene Regulation and Expression, School of Life Sciences, University of Dundee, Dundee DD1 5EH, UK. [2] Centre of Biophotonics, School of Physics and Astronomy, University of St. Andrews, St. Andrews KY16 9SS, UK. [3] Biomedical Sciences Research Complex, School of Biology, University of St. Andrews, St. Andrews KY16 9ST, UK. ✉email: jcp10@st-andrews.ac.uk; r.t.hay@dundee.ac.uk

Ubiquitination is a eukaryotic post-translational modification that modulates a host of cellular processes[1]. Modification is mediated by an E1 activating enzyme (E1), an E2 conjugating enzyme (E2) and an E3 ligase (E3). The E1 catalyses formation of a highly reactive thioester linked conjugate (E2~Ub) between ubiquitin and E2[2]. The largest class of ubiquitin E3 ligases, which is represented by RING E3s, bind both substrate and E2~Ub and facilitate transfer of ubiquitin from the E2 to substrate. Crystal structures of stable E2~Ub conjugates in complex with RING E3 ligases have revealed that the E2~Ub conjugate binds to RING E3 ligases in a closed conformation[3–6], with some requiring RING domain dimerisation[7,8], while others require a non-RING element such as phosphorylation or an additional ubiquitin[9,10], to stabilise the closed conformation. The closed conformation has also been observed for other RING E3 catalysed reactions involving Ubls such as SUMO and Nedd8[11,12]. Ubiquitin in an E2~Ub conjugate can exist in multiple conformations and NMR experiments have generated open and closed models of the E2~Ub conjugate alone[13–15] and in complex with a RING/Ubox in solution[16].

Despite the wealth of structural and biochemical data supporting the closed conformation as the active conformation in RING E3 catalysis, it has not been observed directly in a ubiquitination reaction as the conformation from which the E2~Ub conjugate reacts. We therefore decided to use single-molecule FRET to access conformational states of the E2~Ub conjugate undergoing a RING E3 catalysed ubiquitination reaction in real-time. Using Ubc13 as an E2, we design a FRET labelled Ubc13~Ub conjugate and define the high FRET state as the closed conformation using a stable isopeptide linked E2~Ub conjugate, while the low FRET state represents more open conformations. Using RNF4 as a RING E3 ligase, which targets SUMOylated proteins for ubiquitination, and a reactive thioester linked E2~Ub conjugate, our real-time smFRET experiment reveals that the RNF4 catalysed ubiquitination reaction of a SUMOylated substrate proceeds from the high FRET state or closed conformation of the E2~Ub conjugate.

## Results

### RNF4/Ubc13~Ub assembly and ubiquitin transfer requires UEV.
We have previously crystallised Ubc13~Ub in complex with the RNF4 RING domain dimer and the Ubc13 binding partner, UEV, where the Ubc13~Ub conjugate adopts the closed or active conformation with ubiquitin folded back onto the E2, exposing the thioester to nucleophilic attack by K63 of the substrate ubiquitin (Fig. 1a). Crystallisation of the complex without UEV also yielded the same closed conformation of the Ubc13~Ub conjugate[5] (Fig. 1b), even although RNF4 fails to transfer ubiquitin from the Ubc13~Ub conjugate in the absence of UEV (Fig. 1c, d). In the assay, formation of K63–linked ubiquitin chains on the substrate results in an increasing fluorescence polarisation signal, while a low fluorescence polarisation signal is observed in assays lacking UEV (Fig. 1c). In the absence of UEV ubiquitin cannot be transferred to substrate leading to a build up of the thioester linked Ubc13~Ub conjugate as shown by analysis of the reaction products in SDS-PAGE (Fig. 1d, Supplementary Fig. 1). We found that the RNF4 RING domain dimer has a higher binding affinity for the Ubc13~Ub conjugate when in complex with UEV (Fig. 1e) suggesting that UEV plays a central role in priming of the Ubc13~Ub conjugate for catalysis when bound to the dimeric RING of RNF4. Crystal structures in the presence and absence of UEV show complex formation between the RNF4 RING domain dimer and the Ubc13~Ub conjugate in the closed conformation, which is believed to represent the active conformation of the E2~Ub conjugate. However, the high concentration of protein

used for crystallisation of the complex, may allow the Ubc13~Ub conjugate to adopt the closed conformation, even although it is incapable of ubiquitin transfer. We therefore decided to use smFRET to interrogate the structure of the Ubc13~Ub conjugate alone and in response to RNF4 and UEV binding, and RING E3 catalysed ubiquitin transfer.

### UEV and RNF4 stabilise the closed conformation of Ubc13~Ub.
Single molecule fluorescence approaches have been applied to the ubiquitin system[17–19]; but not to follow RING catalysed transfer of ubiquitin from E2 to substrate. We thus designed a Ubc13~Ub conjugate labelled with the Cy3B-AlexaFluor 647 FRET pair ($R_0 = 60$ Å) where proximity of the FRET labels in the closed conformation yields a high FRET efficiency, while the large distance between the labels in the more open conformation results in a low FRET efficiency (Fig. 2a, Supplementary Fig. 2a, b). The closed and open conformations represent the shortest and longest distance between the FRET labels respectively, however, occupation of ubiquitin in the space between a closed and open conformation of ubiquitin with respect to the E2, may produce an intermediate FRET efficiency. We used a stable isopeptide linked Ubc13~Ub conjugate[3,5] to first determine the FRET state of the conjugate in solution and the influence that UEV and the RNF4 RING domain dimer have on its conformation. The dye-labelled conjugate containing his-tagged ubiquitin bound with similar efficiency to the RNF4 RING domain dimer as the untagged and unlabelled conjugate (Fig. 1e, Supplementary Fig. 2c).

The Ubc13~Ub conjugate alone accesses a wide range of FRET states at low temperature (12 °C) and exhibits a slight preference (41%) for a high FRET state with a FRET efficiency value ($E_{FRET}$) of 0.71, indicating that the closed conformation is favoured (Supplementary Figs. 3 and 4a). At higher temperatures (22 and 35 °C), an intermediate FRET state ($E_{FRET}$ ~0.57) and a low FRET state ($E_{FRET}$ ~ 0.36) are favoured and no high FRET state was observed (Fig. 2b, Supplementary Fig. 4b, c). At all temperatures, in the presence of UEV, the low FRET population is stabilised while the intermediate and high FRET states are destabilised, indicating a more open conformation of the conjugate in complex with UEV (Fig. 2b, Supplementary Figs. 3 and 4a–c). At all temperatures, addition of the RNF4 RING domain dimer to the Ubc13~Ub conjugate in complex with UEV, captures the high FRET state, consistent with stabilisation of a closed or active conformation of the Ubc13~Ub conjugate (Fig. 2b, Supplementary Figs. 3 and 4a–c). In the absence of UEV, the RNF4 RING domain dimer does not stabilise the high FRET state at 22 °C (Fig. 2b, Supplementary Fig. 4b), showing that a preassembled UEV/Ubc13~Ub complex is required for the RING dimer to capture the closed conformation. Most single-molecule trajectories displayed the same FRET state until photobleaching occurred and only 1% of them showed rare interconversion events between FRET states within the time window of the measurement (~1 min) (Fig. 2c, Supplementary Fig. 5a–c). Cross-correlation analysis of the predominant single-state trajectories confirmed the absence of fast hidden dynamics within our time resolution of 100 ms for each of the three FRET states observed (Supplementary Fig. 6a–c). Taken together, these data highlight the long-lived stability of each of the complexes present in solution at equilibrium conditions. To establish that changes in FRET observed in the Ubc13~Ub conjugate after addition of UEV and RNF4, were a consequence of UEV and RNF4 binding, we analysed a series of mutations that disrupt UEV-E2 and RNF4-E2~Ub interactions and reduce ubiquitination activity (Supplementary Fig. 7a–d). Single-molecule FRET histograms at 22 °C of the Ubc13~Ub conjugate alone (Fig. 2b), or in the presence of

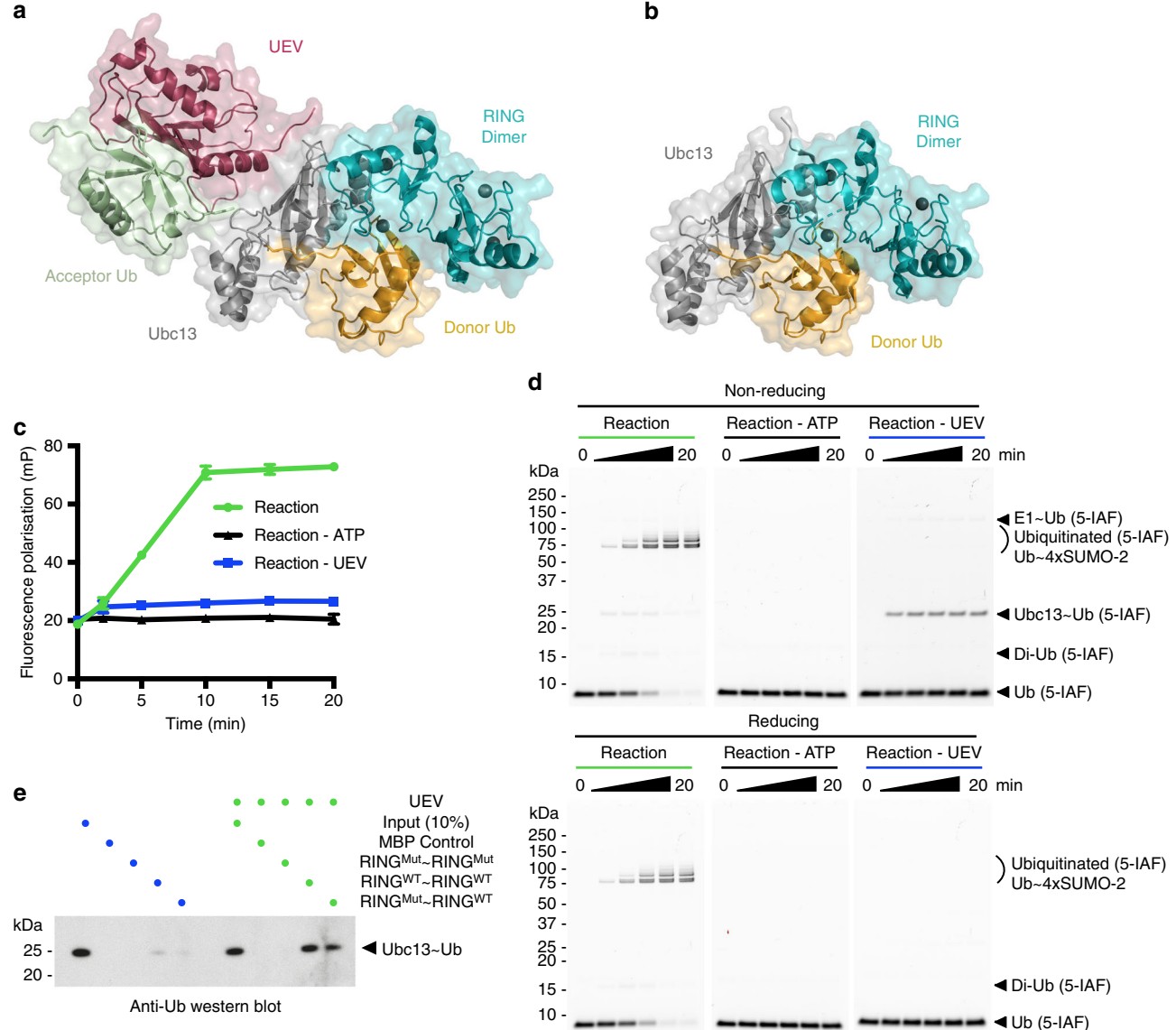

**Fig. 1 RNF4/Ubc13-Ub assembly and ubiquitin transfer requires UEV. a** Structural model of the RNF4 RING domain dimer (cyan) in complex with the Ubc13-Ub conjugate (Ubc13 in grey, Ub in yellow), UEV (raspberry) and acceptor ubiquitin (green) (PDB 5AIT). **b** Crystal structure of the RNF4 RING domain dimer in complex with the Ubc13~Ub conjugate alone (PDB 5AIU). For **a**, **b**, binding of only one Ubc13-Ub conjugate is displayed for clarity. **c** Fluorescence polarisation ubiquitination assay showing the effect of UEV on K63-linked polyubiquitin chain formation using fluorescein labelled version of Ubiquitin, Ub (5-IAF). Data points represent the mean of $n = 3$ independent experiments and error bars represent ±s.d. Data points are represented for the Reaction by green circles, for the Reaction - ATP by black triangles and for the Reaction - UEV by blue squares, with a line drawn in the same colour through each data point in a reaction. Error bars are omitted when the error is smaller than the data point. **d** SDS-PAGE analysis of fluorescence polarisation ubiquitination assays shown in **c**, under non-reducing (above) and reducing (below) conditions imaged using fluorescein filter. **e** RNF4 RING domain dimer pull-down experiment showing the effect of UEV on Ubc13-Ub conjugate binding, analysed by western blot. WT denotes a wild-type RING domain, while Mut denotes a RING domain containing E2~Ub binding site mutations (M140A and R181A). Linearly fused RING domain dimers are used to ensure the RING domain dimers of RNF4 are constitutively dimeric. For **d**, **e**, the same colour scheme is used as in **c**. Source data are provided as a Source Data file.

these mutant proteins (Supplementary Fig. 7e), were indistinguishable indicating that changes in smFRET upon addition of wild-type UEV and the RNF4 RING domain dimer were due to direct binding to the Ubc13~Ub conjugate.

Multiple conformations of ubiquitin with respect to Ubc13 may contribute to the intermediate FRET state. As the intermediate FRET state is detected in the presence of UEV (Fig. 2b, Supplementary Figs. 3 and 4a–c), this indicates that the space occupied by UEV can be excluded as a region that isopeptide linked ubiquitin can occupy in the intermediate FRET state (Fig. 1a). Increased temperature is anticipated to destabilise

the high FRET state or closed conformation due to breakage of the hydrogen-bonding network in the active site groove that is required for adopting the closed conformation and drives specificity for catalysis in the active site[3]. As a result the C-terminal tail of ubiquitin will sit outside the active site groove similar to a previous NMR model[13] leading to a sub-optimal conformation. This conformation may give rise to the intermediate FRET state (Fig. 2b, Supplementary Figs. 3 and 4a–c), in particular in the presence of UEV and it is likely that the hydrophobic interaction between ubiquitin and Ubc13 in the E2~Ub conjugate, is still engaged in this conformation.

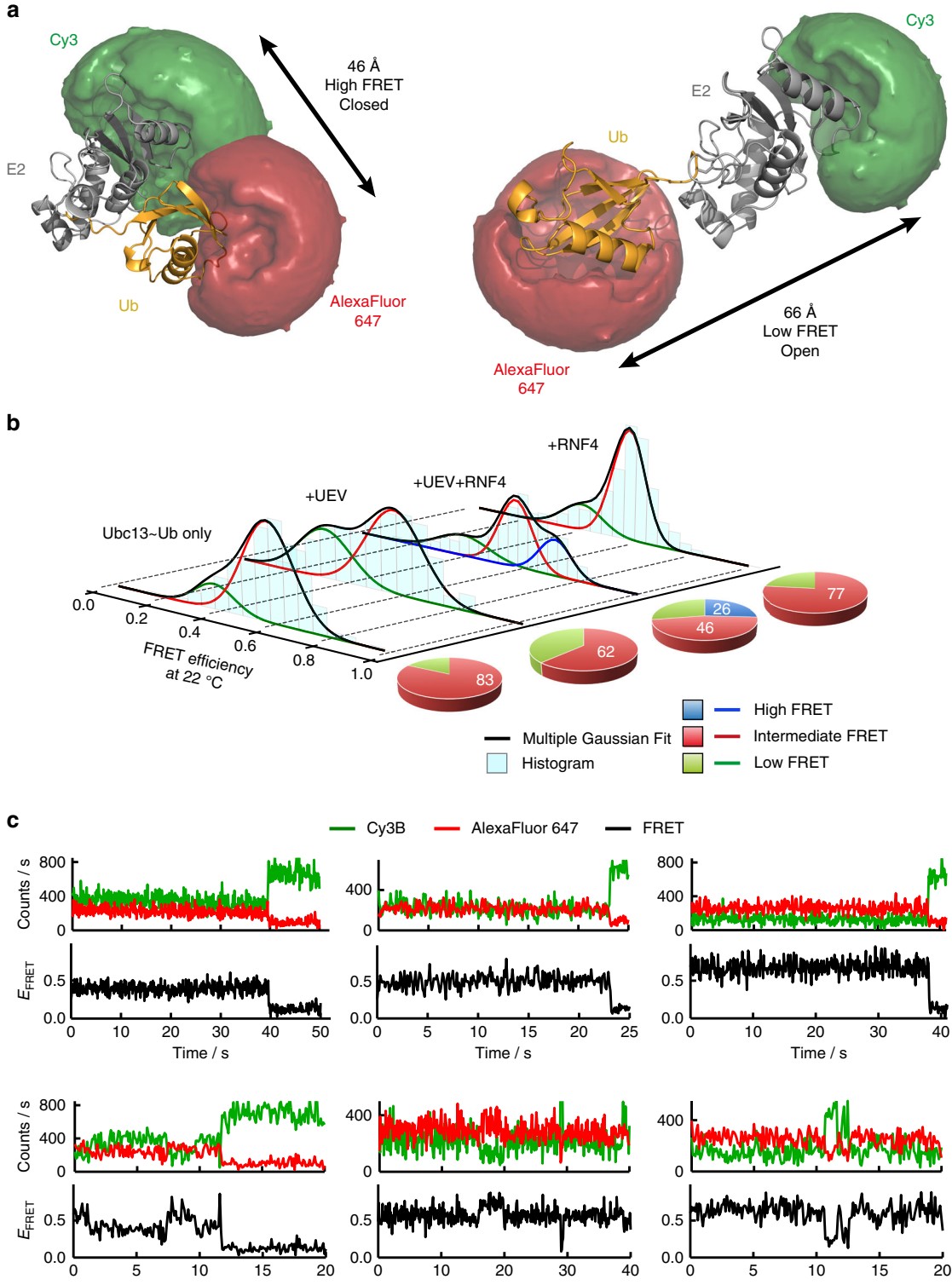

**Fig. 2 UEV and RNF4 stabilise the closed conformation of Ubc13-Ub. a** Modelled accessible volume of FRET dyes (Cy3 in green and AlexaFluor 647 in red) and average distance measurement for E2~Ub conjugate closed and open conformations. **b** smFRET histograms (cyan) showing the isopeptide linked E2~Ub conjugate conformation alone and in complex with UEV and the RNF4 RING domain dimer at 22 °C. The E2~Ub conjugate conformation in the presence of the RNF4 RING domain dimer is also shown. Gaussians are fitted to the low FRET state (green), the intermediate FRET state (red) and the high FRET state (blue). The multiple Gaussian fit is shown in black. The pie charts to the right of each histogram show the percentage contribution of each FRET state to the overall population, using the same colour scheme as the Gaussians fitted to the smFRET histograms. **c** Representative single-molecule traces for the E2~Ub conjugate in complex with UEV and the RNF4 RING domain dimer. Each panel shows the Cy3B donor intensity (green), the AlexaFluor 647 acceptor intensity (red) and the FRET trace (black). Traces demonstrating the stability of the low, intermediate and high FRET states observed for most complexes are shown in the upper panels. Examples of traces within the 1% that show interconversion between FRET states are shown in the lower panels. Source data are provided as a Source Data file.

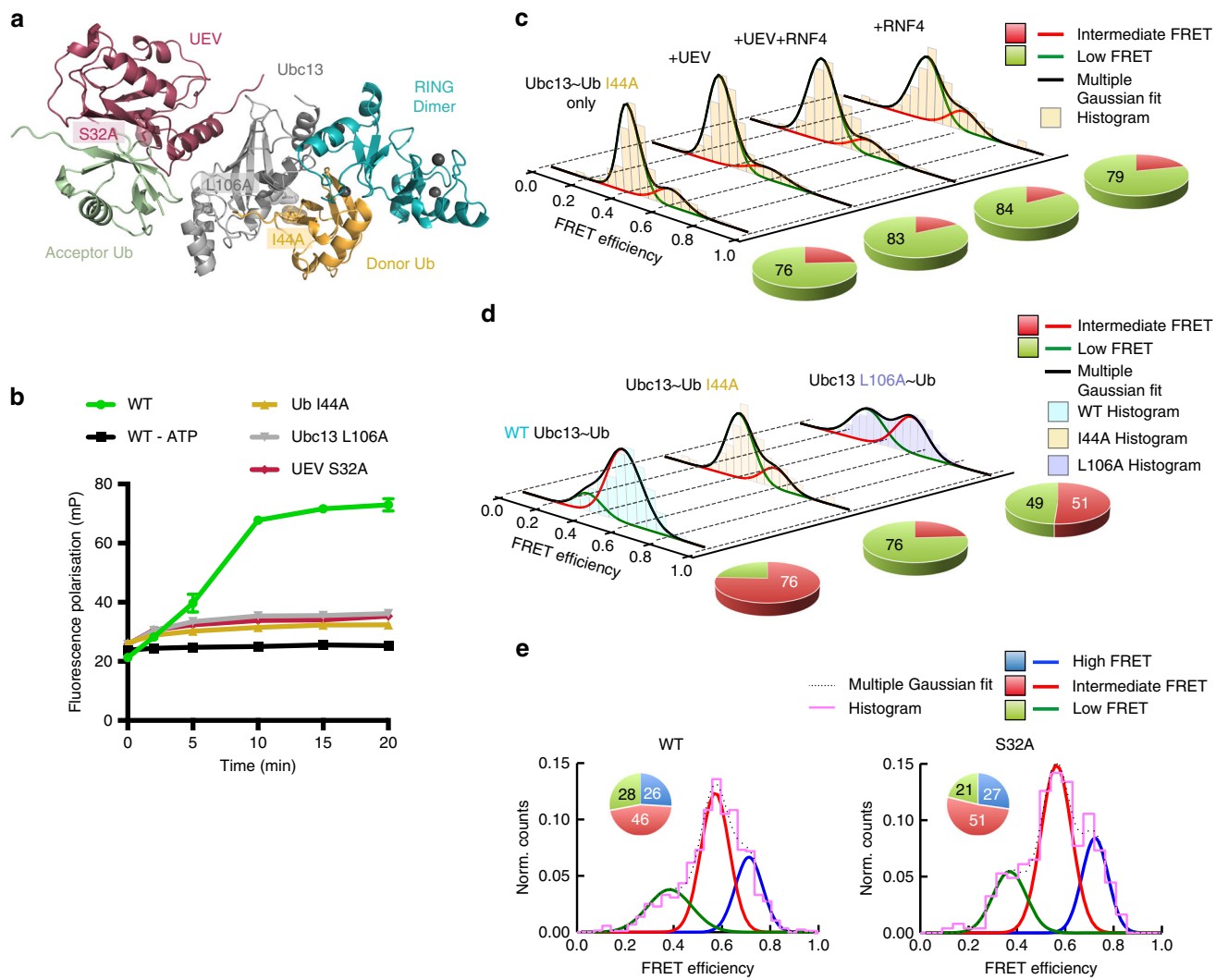

**Fig. 3 The role of specific amino acids in the E2~Ub conformation. a** Locations of mutations made within the complex (PDB accession number 5AIT). **b** Fluorescence polarisation ubiquitination assay showing the effect of mutations on K63-linked polyubiquitin chain formation. Data points represent the mean of $n = 3$ independent experiments and error bars represent ±s.d. Data points are represented for the reaction containing Ub I44A by yellow triangles, Ubc13 L106A by grey inverted triangles and UEV S32A by raspberry diamonds, with a line drawn in the same colour connecting each data point in a reaction. Error bars are omitted when the error is smaller than the data point. **c** smFRET histograms (yellow) showing the E2~Ub conjugate conformation containing Ub I44A alone and in complex with UEV and the RNF4 RING domain dimer. **d** smFRET histograms showing the conformation of the WT E2~Ub conjugate (cyan) and the E2~Ub conjugate containing either Ub I44A (yellow) or Ubc13 L106A (purple) at 22 °C. For **c**, **d**, pie charts to the right of each histogram highlight the percentage contribution of each FRET state to the overall population. **e** Normalised smFRET histograms (magenta) showing conformation of the E2~Ub conjugate in complex with the RNF4 RING domain dimer and either WT UEV (left panel) or UEV S32A (right panel) at 22 °C. The multiple Gaussian fit is represented by a black dotted line. Pie chart insets show the percentage contribution of each FRET state to the overall population. Source data are provided as a Source Data file.

**The role of specific amino acids in the E2~Ub conformation.** To directly test the contribution of hydrophobic contacts in each of the complexes present in solution and confirm that the high FRET state represents the closed conformation we made mutations in the hydrophobic interaction surface and analysed them using smFRET. We used the I44A mutation in ubiquitin (Fig. 3a), which has dramatically reduced ubiquitination activity in vitro (Fig. 3b, Supplementary Fig. 8a, b) and disrupts the closed conformation by reducing the interaction between the hydrophobic patch of ubiquitin and E2. This mutation stabilises the low FRET state indicating a more open conformation and the conjugate cannot access the high FRET state or closed conformation, even in the presence of UEV and the RNF4 RING domain dimer (Fig. 3c, Supplementary Fig. 9a). An L106A mutation in Ubc13

that is part of the same hydrophobic interface as Ub I44A, displays similar ubiquitination defects in vitro (Fig. 3a, b, Supplementary Fig. 8a, b), and has a similar, but less pronounced, effect on the conformation of the Ubc13~Ub conjugate (Fig. 3d, Supplementary Fig. 9b). In addition to destabilising the high FRET state, these mutations also destabilise the intermediate FRET state (Fig. 3d, Supplementary Fig. 9b), indicating that the hydrophobic interaction maintains both the intermediate and high FRET states, and when disrupted a more open conformation or low FRET state is favoured. An S32A mutation in UEV[20] that perturbs binding of the acceptor ubiquitin (Fig. 3a), was defective in RNF4 dependent substrate ubiquitination (Fig. 3b, Supplementary Fig. 8a, b), but retained the ability to activate the Ubc13~Ub conjugate in an RNF4 dependent lysine discharge assay[5]. To

determine if the S32A mutation in UEV could co-operate with RNF4 to restrain the Ubc13~Ub conjugate in the closed conformation, we carried out smFRET analysis of the wild type and S32A versions of UEV. This revealed that the conformation of the Ubc13~Ub conjugates were indistinguishable and indicated that stabilisation of the closed conformation was independent of acceptor ubiquitin binding to UEV (Fig. 3e). These data also indicate that none of the FRET states observed are a consequence of ubiquitin binding to UEV.

**Single ubiquitination events revealed by real-time smFRET.** Use of stable isopeptide linked Ubc13~Ub conjugate allowed us to identify the high FRET state as the closed conformation and indicated that both UEV and RNF4 were required to restrain the Ubc13~Ub conjugate in this conformation. To directly test the hypothesis that the closed conformation is the active form of the Ubc13~Ub conjugate requires a method that reveals the FRET state(s) from which ubiquitin transfer to substrate occurs. This necessitated attaching the same FRET dyes at identical positions used for the isopeptide linked conjugate to an reactive thioester linked conjugate (Supplementary Fig. 10a, b). The labelled conjugate modified the substrate in a biochemical assay with similar efficiency to the WT conjugate (Supplementary Fig. 11). smFRET analysis at 22 °C of the reactive thioester linked E2~Ub conjugate alone revealed that while the thioester has access to all three previously defined FRET states, the high FRET state or closed conformation is more populated in comparison to the stable isopeptide linked E2~Ub conjugate (Supplementary Fig. 12a, b). This may be a consequence of differences in the length and precise geometry of the linkage between ubiquitin and Ubc13 in the isopeptide mimic and the native thioester bonds. In the smFRET experiment we immobilised the unstable thioester linked Ubc13~Ub conjugate onto the surface for imaging and injected the reaction components for ubiquitination including UEV, RNF4 and Ub~4xSUMO-2 substrate. We monitored, in real-time, transfer of Cy3B-labelled ubiquitin from the E2~Ub conjugate onto substrate, which diffuses into solution. This reaction results in simultaneous loss of both donor and acceptor signals as well as complete loss of the FRET signal (Fig. 4a). This is evident from the decrease in number of FRET pairs before and after the reaction has occurred (Fig. 4b). Following the cumulative intensity of fluorescent spots over time produces a stable fluorescent signal with the injection of UEV alone, while the injection of all reaction components including UEV, RNF4 and substrate, results in the loss of fluorescence as a result of RNF4 catalysed ubiquitin transfer to substrate (Fig. 4c). A similar trend is observed for the Cy3B donor signal, while the AlexaFluor 647 acceptor signal experiences a slightly faster loss of signal in the UEV injection in comparison to Cy3B due to background photobleaching events for AlexaFluor 647. Single-molecule trajectories demonstrate stable Cy3B and AlexaFluor 647 intensities lasting hundreds of seconds following a UEV only injection as well as the loss of AlexaFluor 647 due to slow photobleaching events and concomitant Cy3B recovery (Fig. 4d). The ubiquitination reaction is characterised by the rapid simultaneous loss of Cy3B and AlexaFluor 647 as well as the resultant FRET (Fig. 4d), as set out in the experimental design (Fig. 4a). There is no recovery of FRET within the single molecules that have undergone ubiquitin transfer. Combined analysis of single molecule traces shows a significant portion of E2~Ub conjugates react upon injection of reaction components (~60%), compared to injection of UEV only (~14%) (Fig. 4e). A small portion of the sample contains only Cy3B due to under-labelling with AlexaFluor 647 or prior photobleaching of AlexaFluor 647, however, a reaction was also observed in these conjugates (Fig. 4e).

**RNF4 catalyses ubiquitin transfer from a closed conformation.** Contour-plots of FRET trajectories in a 90-s time-window following initiation of the ubiquitination reaction showed the rate of FRET loss due to ubiquitin transfer to substrate depends on RNF4 concentration and FRET loss was not observed in the absence of RNF4 or with an RNF4 mutant unable to bind the Ubc13~Ub conjugate (Fig. 5a). Incubation in the presence of UEV and RNF4 but in the absence of substrate also resulted in no loss of FRET in the contour plot and cumulative intensity of fluorescent spots over time produced a stable fluorescent signal (Supplementary Fig. 13a, b), indicating that ubiquitin is specifically transferred to substrate and is not discharged to any other components of the reaction mixture. Single-molecule dwell-time analysis demonstrated that ubiquitin transfer rates can be distinguished from slow Cy3B and AlexaFluor 647 photobleaching events and quantified via dwell time of the FRET signal (Fig. 5b, c, Supplementary Fig. 13c). Ubiquitin transfer reactions from both Cy3B only and FRET containing conjugates show similar rate in ubiquitin transfer and dependence on RNF4 (Fig. 5c). The smFRET population histogram derived from only those molecules undergoing ubiquitin transfer, accurately shows the RNF4 dependent reaction proceeded from the high FRET state ($E_{FRET}$~0.71) similar to that assigned to the closed conformation of the isopeptide linked E2~Ub conjugate (Fig. 5d). A low fraction of molecules reacting at a slower rate and from a low FRET state were also present in the UEV only injection, indicating that this was a background reaction that is not dependent on RNF4 (Fig. 5e, f). Contour plots of all FRET active molecules show an RNF4 dependent loss of FRET from the high FRET state only, while the middle FRET state is almost constant over time and the low FRET state is inactive (Supplementary Fig. 14a–c). This is further demonstrated by histograms generated from 15-s intervals showing the loss of only the high FRET state population in the RNF4 catalysed reaction (Supplementary Fig. 14a). In contrast, the distribution of FRET populations observed for the RNF4 mutant remained unaltered (Supplementary Fig. 14b) even 6 min after injection, thus confirming the lack of interconversion between states and the extremely long-lived stability of the different complexes formed (Supplementary Fig. 15a, b). Stabilisation of the high FRET state in the absence of substrate indicates RNF4 binding but lack of ubiquitin transfer to substrate (Supplementary Fig. 14c).

**Discussion**
Previous crystallographic studies of RING E3 ligases bound to E2~Ub conjugates, show the conjugate restrained in the closed conformation[3,4]. Further crystallographic studies of RING bound E2~Ub conjugates with the addition of substrate, show the E2~Ub conjugates in the same closed conformation with the substrate lysine aligned to attack the thioester bond[5,11]. These structural studies have employed a range of chemical biology approaches to stabilise suitable complexes for structural analysis. Although yielding high-resolution structural information, these structures represent a snapshot of the arrangement of proteins prior to the catalytic step. More recent studies have chemically stabilised substrates in the active site of RING bound E2~Ub conjugates; in an effort to mimic the transition state of a ubiquitin transfer reaction[12,21]. These structures also show stabilisation of the E2~Ub conjugate in a closed conformation. Stabilisation of E2~Ub conjugate in complex with RING E3 ligases yields high-resolution structural gains, however structural information about the ubiquitination reaction in real time is lost. NMR studies[13–16] provided structural information on the conformational states of the E2~Ub conjugate, however such ensemble measurements require hours of averaging to generate a model. Mutational and

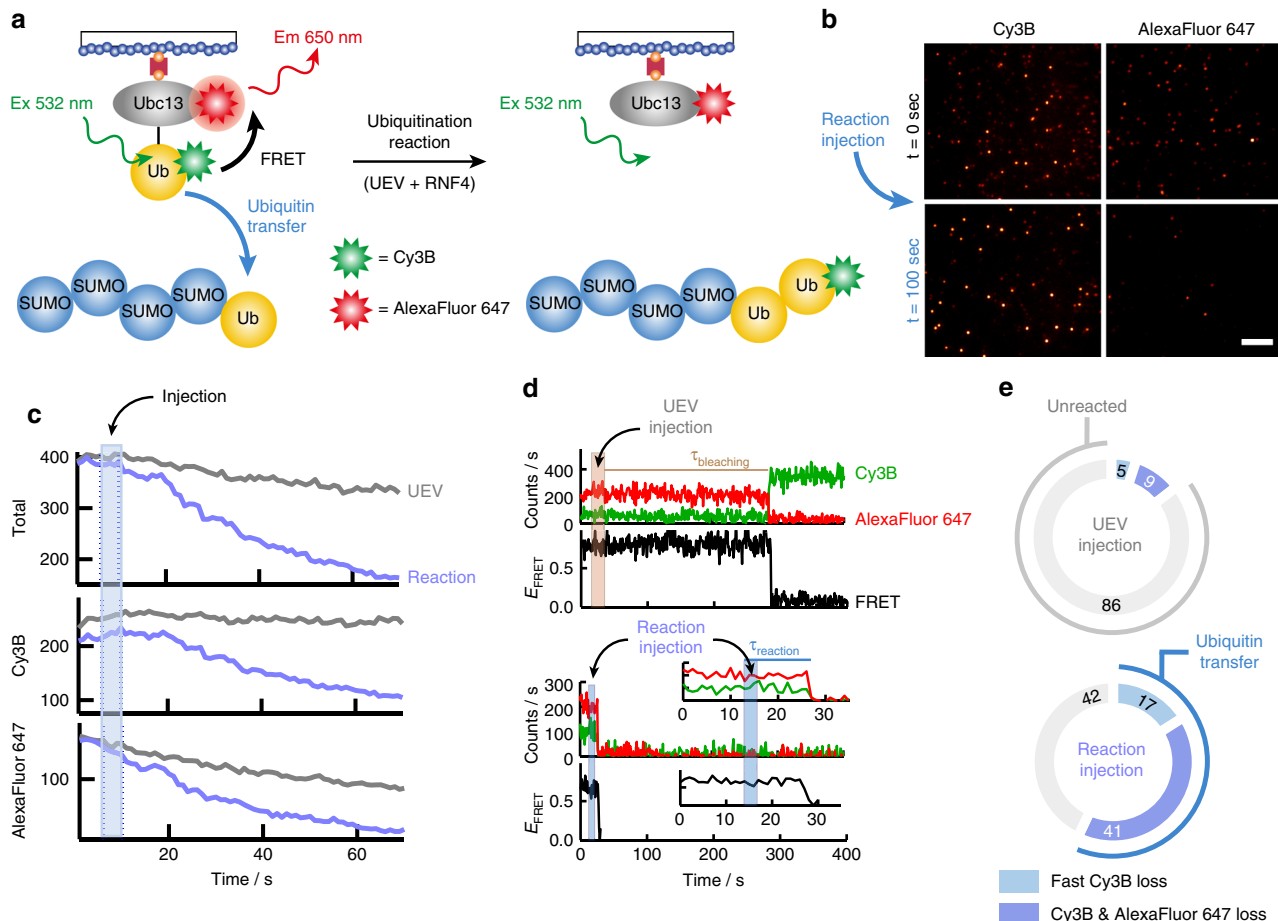

**Fig. 4 Single ubiquitination events revealed by real-time smFRET. a** Schematic diagram showing real-time ubiquitination reaction results in loss of FRET between the Ubc13~Ub conjugate and transfer of Cy3B-labelled ubiquitin on to the substrate that diffuses into solution. Excitation (Ex) and emission (Em) wavelengths are highlighted. **b** Images of Cy3B and AlexaFluor 647 channels taken before (t = 0 s, black) and after (t = 100 s, blue) the injection of reactants showing loss of both the Cy3B signal and the FRET signal from AlexaFluor 647 during the real-time smFRET experiment. The scale bar represents 10 μm. **c** Comparison of cumulative intensity variation from all fluorescent spots over time for UEV only injection (grey) versus reaction injection (blue). The total fluorescence intensity is shown along with separate Cy3B and AlexaFluor 647 intensities, with the injection interval highlighted in blue. **d** Comparison of representative single molecule traces for the UEV injection (top) and reaction injection (bottom). Each molecule contains a Cy3B, AlexaFluor 647 and FRET intensity trace. The injection interval for the UEV injection (brown) and the reaction injection (blue) is highlighted. The lifetime of AlexaFluor 647 before photobleaching ($\tau_{bleaching}$) is highlighted in brown for the UEV injection. The inset shows the first 35 s of the reaction injection, highlighting in blue the simultaneous loss and short lifetimes ($\tau_{reaction}$) of the Cy3B, AlexaFluor 647 and FRET signals. **e** Charts showing the percentage contribution of unreacted molecules (grey) and molecules undergoing ubiquitin transfer (Cy3B loss in light blue, Cy3B and AlexaFluor 647 loss in dark blue) to the overall population, representing n = 263 and 270 individual molecules for the UEV and reaction injection respectively. Source data are provided as a Source Data file.

biochemical studies have provided insight into the requirement of the interaction surfaces observed in these complexes in ubiquitin transfer reactions[3–5,11,12,16]. However, the structure of the E2~Ub conjugate has not been directly studied in real-time with a rapid structural output during a RING E3 catalysed ubiquitin transfer reaction, to determine the catalytically active conformation. Similarly, stopped flow biochemical analysis[22] generates rapid kinetics, but it lacks structural information.

In this study, smFRET provided access to the conformation of individual E2~Ub conjugates in real-time undergoing rapid substrate ubiquitination. We benefited from the wealth of high-resolution structural information available, from which we generated models of open and closed forms of the E2~Ub conjugate. smFRET analysis of the stable isopeptide linked E2~Ub conjugate yielded information about the conformational space available in solution, which includes a variety of closed, open and intermediate conformations. In addition, we directly tested the hypothesis that the closed conformation represented the active

conformation by monitoring the conformation of the native and reactive thioester linked E2~Ub conjugate in a real-time RING E3 catalysed ubiquitination reaction. Without any chemical modifications to the active site we showed that RING E3 catalysed ubiquitin transfer to substrate occurs from a closed conformation of the E2~Ub conjugate.

In the real-time ubiquitination reaction we found that RNF4, UEV and a suitable substrate are required to promote RING E3 catalysed ubiquitin transfer to substrate from a closed conformation of the Ubc13~Ub conjugate. In the absence of any of these three components, ubiquitin transfer did not occur, even though a closed conformation was accessible to the Ubc13~Ub conjugate. In essence, while the RING E3 catalysed ubiquitination reaction occurs from the closed conformation, the closed conformation is not the only requirement for ubiquitin transfer. In addition to RING E3 catalysed ubiquitination, other E2~Ubls containing SUMO and Nedd8[11,12] are similarly restrained in the closed conformation by their cognate RING E3 ligases, indicating

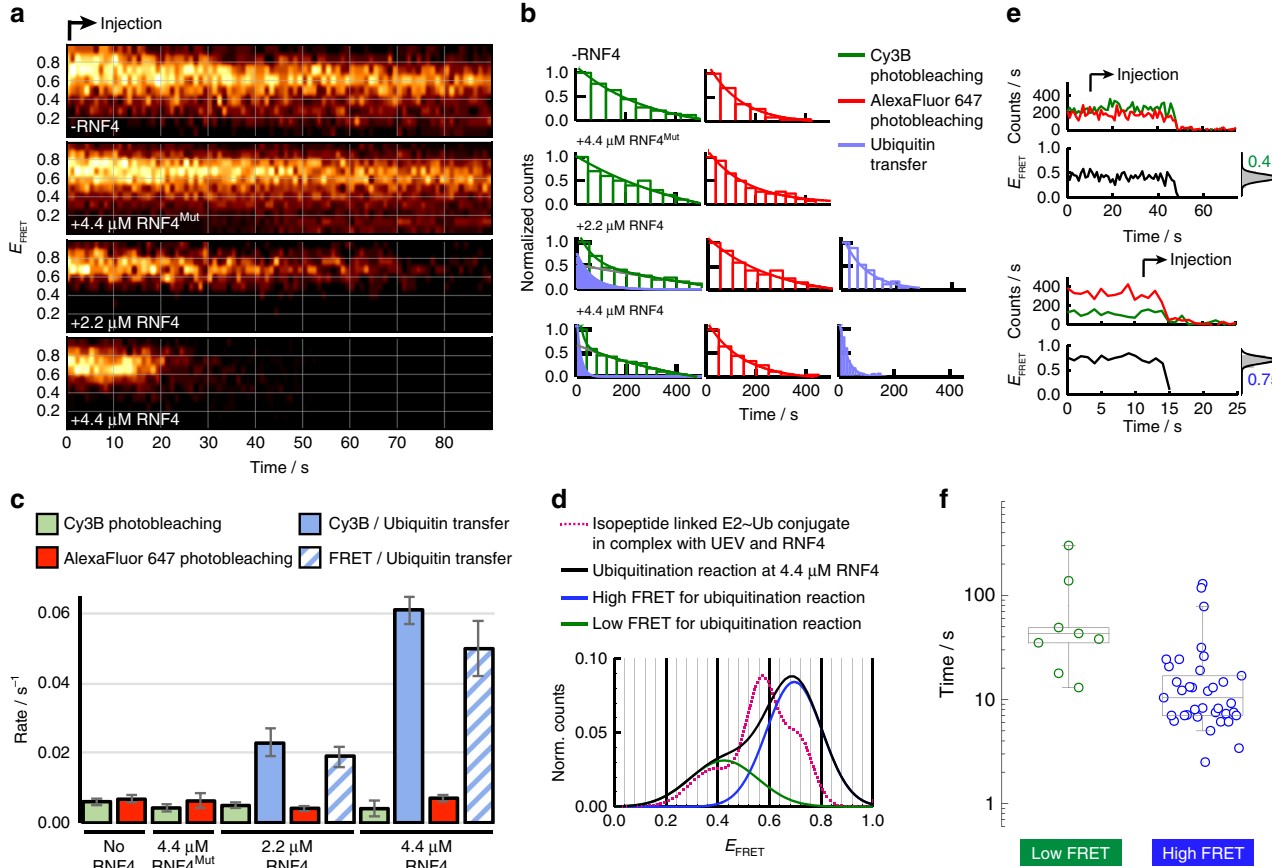

**Fig. 5 RNF4 catalyses ubiquitin transfer from a closed conformation. a** Single molecule contour plots showing the progression of the FRET trajectory following the real-time injection of reactants at the indicated experimental conditions and concentrations. Mut denotes the RNF4 RING domain dimer containing E2~Ub binding site mutations (M140A and R181A). **b** smFRET dwell-time histograms obtained for Cy3B photobleaching (green, left panel), fast Cy3B loss (blue, left panel), AlexaFluor 647 photobleaching (red, middle panel) and FRET loss events (blue, right panel). The solid lines represent the fit to single exponential decay functions. **c** Comparison of the rates of Cy3B (green) and AlexaFluor 647 (red) photobleaching and ubiquitin transfer, obtained by fitting the dwell-time histograms in **b** to mono- or biexponential decay functions. Ubiquitin transfer is split into the rate of loss of the FRET signal (blue stripe) and the loss of Cy3B signal from molecules containing Cy3B only (solid blue). Error bars represent the standard error of the rate extracted from non-linear least squares fitting of the histograms in **b**. **d** Gaussian profile of relative FRET populations for the stable isopeptide linked conjugate (magenta dotted line) and the real-time ubiquitination reaction. The histograms were omitted for clarity. **e** Representative traces for reactions from low (top) and high (bottom) FRET states. Injection start time is indicated along with corresponding FRET histogram (grey). **f** Box-plot comparison of rate of reaction at 4.4 μM RNF4 from the low (green circles, $n = 8$ molecules) and high FRET (blue circles, $n = 36$ molecules) states. The extremes, upper and lower quartiles of the distribution, and the median are represented by the whiskers, box and middle lines, respectively. Source data are provided as a Source Data file.

that the reaction from the closed conformation is likely to be a universal mechanism for RING E3 catalysis.

## Methods
**Cloning, expression and purification of recombinant protein**. Human Ubc13 (also known as Ube2N) was previously subcloned into the pHISTEV30a vector using NcoI and HindIII restriction sites[5]. A sequence encoding an avitag (Gly-Leu-Asn-Asp-Ile-Phe-Glu-Ala-Gln-Lys-Ile-Glu-Trp-His-Glu) followed by a linker (Gly-Gly-Ala) was inserted between the TEV cleavage site and Ubc13 using the NcoI restriction site. A K24C mutation was introduced into Ubc13 using site-directed mutagenesis. Additional C87K and K92A mutations in Ubc13 were used to generate the stable isopeptide linked Ubc13~Ub conjugate[5]. Ubc13 variants were expressed in *E. coli* BL21 (DE3) cells and UEV variants with an N-terminal his₆ and MPB tag were expressed in *E. coli* Rosetta 2 (DE3) cells[5]. Ub~4xSUMO-2 was expressed with an N-terminal his₆ tag in *E. coli* Rosetta 2 (DE3) ElaD knockout cell line[23]. Expression of Ubc13 and UEV variants, and Ub~4xSUMO-2 was induced with 100 μM IPTG at 20 °C overnight. All cells were pelleted and resuspended in 50 mM Tris pH 7.5, 250 mM NaCl, 10 mM Imidazole and Complete Protease Inhibitor Cocktail (EDTA free, Roche) and lysed by sonication. Expression of RNF4 and linear fusions of RNF4 was performed as previously described[7]. Briefly, RNF4 constructs containing an N-terminal his₆ and MBP tag were expressed in *E. coli* Rosetta 2 (DE3) cells with 100 μM IPTG at 20 °C overnight. Pelleted cells were resuspended in 50 mM Tris pH 7.5, 0.5 M NaCl, 10 mM Imidazole and Complete Protease Inhibitor Cocktail (EDTA free, Roche) and lysed by sonication. His₆

tagged Ubiquitin M-2C was previously cloned into a pHISTEV30a vector encoding four extra N-terminal residues (Gly-Ala-Cys-Gly) prior to the native ubiquitin N-terminus[3,24]. An additional K63R mutation in ubiquitin was used to generate the reactive thioester linked Ubc13~Ub conjugate. Ubiquitin constructs were expressed in *E. coli* BL21 (DE3) cells with 1 mM IPTG for 4 h at 37 °C. Pelleted cells were resuspended in 50 mM Tris pH 7.5, 250 mM NaCl, 10 mM Imidazole and Complete Protease Inhibitor Cocktail (EDTA free, Roche) and lysed by sonication. All proteins were purified by Ni-NTA chromatography and eluted with 150 mM imidazole, followed by cleavage with TEV protease and Ni-NTA chromatography to remove any uncleaved protein, free his₆ tag and his₆ tagged TEV. All proteins were purified by size exclusion chromatography on a HiLoad Superdex 75 column (GE Healthcare) using 50 mM Tris pH 7.5, 150 mM NaCl and 0.5 mM TCEP as running buffer. As a result of cloning, Ubc13 has four extra residues (Gly-Ala-Met-Ser) before the N-terminal avitag after cleavage with TEV protease.

**Preparation of biotinylated and dye-labelled protein**. Avitagged Ubc13 was biotinylated in a reaction containing 10 mM Tris pH 7.5, 5 mM MgCl₂, 200 mM KCl, 2.5 mM ATP, 0.5 mM d-biotin, 100 μM avitagged Ubc13 and 16 μM BirA and incubated at 20 °C for 4 h followed by 4 °C overnight as previously described[25]. Reaction products were purified by size exclusion chromatography with a HiLoad Superdex 75 16/600 column and 50 mM Tris, 150 mM NaCl, and 0.5 mM TCEP, pH 7.5 as running buffer. Biotinylated Ubc13 K24C C87K K92A was buffer exchanged into 50 mM Tris, 150 mM NaCl, pH 7.0 using a Centri Pure Zetadex-25 gel filtration column (Generon) and labelled with Cy3B maleimide (GE Healthcare) at room temperature for 2 h using a five times molar excess of Cy3B. Excess dye

was removed using a Centri Pure Zetadex-25 gel filtration column and 50 mM Tris, 150 mM NaCl, and 0.5 mM TCEP, pH 7.5 as running buffer. His$_6$ tagged Ubiquitin M-2C was also labelled using the same protocol but with Cy3B maleimide and AlexaFluor 647 C2 maleimide (Thermo Fisher Scientific) separately. All buffers were degassed, all dye labelling reactions and dye-labelled proteins were protected from light and all products were analysed by intact LC-MS.

**Preparation of isopeptide linked Ubc13-Ub conjugate**. An isopeptide linkage between biotinylated and Cy3B-labelled Ubc13 K24C C87K K92A and AlexaFluor 647 labelled his$_6$ tagged Ubiquitin M-2C was generated using a similar method as described previously[3,5]. The conjugation reaction containing 50 μM Ubc13, 60 μM Ubiquitin, 0.8 μM His$_6$-Ube1, 3 mM ATP, 5 mM MgCl$_2$, 50 mM Tris pH 10.0, 150 mM NaCl and 0.5 mM TCEP was incubated at 37 °C for ~21 h. Reaction products were fractionated by Ni-NTA chromatography to isolate the isopeptide linked Ubc13~Ub conjugate and unreacted Ubiquitin, which are both his$_6$ tagged. All buffers were degassed and all dye-labelled proteins were protected from light. Production and purification of the isopeptide linked Ubc13~Ub conjugate was analysed by SDS-PAGE and the FRET labelled protein was imaged using Chemi-Doc MP (Bio-Rad) with Cy3 and AlexaFluor 647 filter settings. Subsequent Coomassie blue staining allowed imaging of all proteins using the Coomassie filter setting. Ubiquitin I44A and Ubc13 L106A containing conjugates were prepared using the same protocol.

**Preparation of thioester linked Ubc13-Ub conjugate**. A thioester linkage between biotinylated Ubc13 K24C and Cy3B-labelled his$_6$ tagged Ubiquitin M-2C K63R was generated in a reaction containing 50 μM Ubc13, 50 μM Ubiquitin, 0.1 μM His$_6$-Ube1, 3 mM ATP, 5 mM MgCl$_2$, 50 mM Tris pH 7.5, 150 mM NaCl, 0.5 mM TCEP and 0.1% (v/v) NP-40 Alternative that was incubated at 37 °C for 20 min. Reaction products were buffer exchanged into 50 mM Tris, 150 mM NaCl, pH 7.0 using a Centri Pure Zetadex-25 gel filtration column. The thioester was labelled on K24C of biotinylated Ubc13 with a five times excess of AlexaFluor 647 C2 maleimide at room temperature for 2 h. Excess dye was removed using a Centri Pure Zetadex-25 gel filtration column and 50 mM Tris and 150 mM NaCl, pH 7.0 as running buffer. All buffers were degassed and all dye labelling reactions and dye-labelled proteins were protected from light. Production and labelling of the thioester linked Ubc13~Ub conjugate was analysed by SDS-PAGE and the FRET labelled protein was imaged using ChemiDoc MP with Cy3 and AlexaFluor 647 filter settings. Subsequent Coomassie blue staining allowed imaging of all proteins in the samples using the Coomassie filter setting.

**Ubiquitination assays**. Fluorescence polarisation ubiquitination assays were performed and analysed as described previously using a fluorescein labelled version of Ubiquitin M-2C (Ub-5IAF)[24]. Briefly, 2.5 μM Ubc13, 2.5 μM UEV, 9.6 μM WT Ubiquitin, 0.4 μM Ub-5IAF, 0.1 μM His$_6$-Ube1, 0.55 μM RNF4, 5.5 μM Ub~4x-SUMO-2, 5 mM MgCl$_2$, 50 mM Tris pH 7.5, 150 mM NaCl, 0.1% (v/v) NP-40 Alternative, 0.5 mM TCEP were incubated at 20 °C for 20 min taking samples at 0, 2, 5, 10, 15 and 20 min. 3 mM ATP was added after the 0-min time point to start the reaction. The reaction was stopped at each time point by mixing it with stopping buffer (50 mM Tris pH 7.5, 150 mM NaCl, 0.5 mM TCEP, 0.1% (v/v) NP-40 Alternative, 150 mM EDTA pH 8.0) in a 2:1 ratio. Fluorescence polarisation was measured using a PHERAstar FS microplate reader with 485 nm excitation and 520 nm emission wavelengths. The assays were resolved by SDS-PAGE under non-reducing and reducing conditions and Ub-5IAF was imaged using ChemiDoc MP with the fluorescein filter setting. Subsequent Coomassie blue staining allowed imaging of all proteins in the assay using the Coomassie filter setting.

The substrate single turnover assay to validate labelled proteins was performed as previously described[7]. In all, 15 μM FRET labelled thioester linked Ubc13~Ub conjugate was mixed in a 1:1 ratio with 15 μM UEV, 1.1 μM RNF4, 11 μM Ub~4xSUMO-2, 50 mM Tris pH 7.5, 150 mM NaCl, 0.1% (v/v) NP-40 Alternative, 0.5 mM TCEP and the reaction incubated at 20 °C for 10 min and stopped with non-reducing SDS-PAGE loading buffer. Samples were taken at 0.5, 1, 2, 5 and 10 minutes. A WT Ubc13~Ub thioester linked conjugate, prepared using the same protocol as described above, was used as a control in this experiment. Ubiquitin in both Ubc13~Ub conjugates contained a K63R mutation such that only one ubiquitin was transferred onto the substrate. An RNF4 construct containing full length RNF4 linearly fused to a second RING domain was used in the ubiquitination reaction, similar to real-time smFRET experiments. Full-length RNF4 includes four N-terminal SUMO Interaction Motifs, which engage the Ub~4xSUMO-2 substrate, and a C-terminal RING domain. One of the E2~Ub binding sites within the RING domain dimer was mutated (M140A and R181A) allowing only one E2~Ub conjugate to bind during the ubiquitination reaction. The assay was resolved by SDS-PAGE and the FRET labelled protein was imaged using ChemiDoc MP with Cy3 and AlexaFluor 647 filter settings. Subsequent Coomassie blue staining allowed imaging of all proteins in the assay using the Coomassie filter setting.

**Pull-down assay**. Binding between an MBP tagged linearly fused dimer of the RING domain of RNF4 and the Ubc13~Ub conjugate was performed as described previously[7]. Production of the linearly fused dimer of the RING domain of RNF4 was described previously[3,7] and was used in the pull-down assay to ensure that the

RING domain is constitutively dimeric, which is required for binding of the Ubc13~Ub conjugate. E2~Ub binding site mutations (M140A and R181A) are used in a single RING domain or both RING domains to disrupt one or both E2~Ub binding sites respectively. In the pull-down experiment, Ubc13~Ub conjugates were mixed at room temperature with His$_6$ MBP tagged RNF4 variants bound to 10 μl amylose beads in 50 mM Tris pH 7.5, 150 mM NaCl, 0.5 mM TCEP, 5% (v/v) glycerol. The amylose beads were washed by resuspension in 0.5 ml of the same buffer. The beads were gently pelleted, followed by aspiration of the supernatant. In the final washing step the beads were resuspended again in 0.5 ml of the same buffer and placed on 1.4 ml of 50 mM Tris pH 7.5, 150 mM NaCl, 0.5 mM TCEP, 5% (v/v) glycerol and 10% (w/v) sucrose. The beads were quickly and gently pelleted and the supernatant was aspired. The remaining proteins bound to the beads were eluted with SDS-PAGE loading buffer and resolved by SDS-PAGE. The results were analysed by western blotting with anti-ubiquitin primary antibody (Dilution 1:2000, Z0458, Dako) and anti-rabbit HRP secondary antibody (Dilution 1:10000, A6154, Sigma Aldrich) for the unlabeled Ubc13~Ub conjugate. The FRET labelled Ubc13~Ub conjugate was analysed by SDS-PAGE and imaged using ChemiDoc MP with Cy3 and AlexaFluor 647 filter settings. Subsequent Coomassie blue staining allowed imaging of the proteins using the Coomassie filter setting. All uncropped gels and blots are provided in the Source Data file.

**FRET dye modelling**. Cy3 and AlexaFluor 647 were modelled on to the E2~Ub conjugate using the FRET-restrained positioning and screening (FPS) software[26]. As Cy3B maleimide is not available in the FPS software package, the closest alternative, Cy3 maleimide, was used for modelling purposes. Ubc13 (Chain B) and Ubiquitin (Chain C) in PDB 5AIT were used for modelling the accessible volume of FRET dyes as well as the distance between them for the closed conformation of the E2~Ub conjugate, while the NMR model (PDB accession number 2KJH) was used for the open conformation of the conjugate. As the cysteine in ubiquitin is contained within an N-terminal linker that is not present in these PDB files, the closest surface accessible amino acid, Gln2, was used for attachment and modelling the dye on ubiquitin. This is likely to result in a distance measurement that is shorter than expected; therefore, the resulting distance calculations were used as a guide.

**Slide passivation**. Slides were passivated as described previously[27]. Aminosilane treated slides and coverslips were passivated for 1 to 3 h with a mixture of biotin-PEG-SVA and PEG-SVA in a mass ratio of 1:100 respectively, dissolved in 100 mM sodium bicarbonate pH 8.5. Slides and coverslips were rinsed well with methanol and water and were subsequently assembled into four channels for stable isopeptide linked conjugate studies. Single channel slides with tubing and syringe attached for injection purposes were assembled for real-time reaction studies using the unstable thioester linked conjugate. Channels were coated with 0.2 mg ml$^{-1}$ neutravidin for 10 min prior to addition of biotinylated FRET labelled proteins.

**Single molecule total internal reflection**. smFRET experiments were performed as described previously[28]. All experiments were performed on a prism-type total-internal reflection microscope using an inverted microscope (Olympus IX71). A 532-nm laser (Crystalaser) was used for Cy3B excitation and images were collected on a back illuminated Ixon EMCCD camera (Andor, 512×512 pixels). Cy3B and AlexaFluor 647 fluorescence were split by dichroic mirrors (DCRLP645, Chroma Technology) into two channels allowing simultaneous imaging with Cy3B on the left and AlexaFluor 647 on the right of the EMCCD camera.

**Isopeptide linked conjugate smFRET experiments**. In total, 50 pM of the stable isopeptide linked FRET labelled Ubc13~Ub conjugate was bound to a slide passivated with biotinylated PEG and neutravidin for 10 min. Excess free AlexaFluor 647 labelled ubiquitin was washed from the surface using 50 mM Tris, 150 mM NaCl, 0.5 mM TCEP, pH 7.5. Imaging buffer containing 50 mM Tris, 150 mM NaCl, 0.5 mM TCEP pH 7.5, 1 mM Trolox, 6.25 mM 3,4-protocatechuic acid (PCA) and 250 nM protocatechuate dioxygenase (PCD) was then added to the slide. The Ubc13~Ub conjugate was imaged alone and in complex with UEV and the RING domain dimer of RNF4. One of the E2~Ub binding sites within the RING domain dimer was mutated (M140A and R181A) allowing only one Ubc13~Ub conjugate to bind to the RING domain dimer. For complex imaging, imaging buffer was supplemented with 10 μM UEV and/or 40 μM RING domain dimer of RNF4. A series of smFRET trajectories were acquired using 100 ms integration time and within ~30 min of the addition of imaging buffer. Experiments for the WT Ubc13~Ub stable isopeptide linked conjugate were performed at three different temperatures including 12, 22 and 35 °C, while experiments for mutant forms of the Ubc13~Ub stable isopeptide linked conjugate and UEV were performed at 22 °C.

**Thioester linked conjugate real-time smFRET experiments**. A flow cell was generated as described previously[29]. Briefly, tubing was attached to the drilled holes at each end of the single channel slide. One piece of tubing was placed in an eppendorf containing the desired solution while a syringe was attached to the other piece of tubing through which the solution was drawn through the channel by suction. Using this suction method, 100 pM of the unstable thioester linked FRET

labelled Ubc13~Ub conjugate was bound to a slide passivated with biotinylated PEG and neutravidin for 10 min. Excess free Cy3B-labelled ubiquitin was washed from the surface using 50 mM Tris, 150 mM NaCl, pH 7.0. Free AlexaFluor 647 labelled Ubc13 is not observed due to direct excitation of Cy3B. Prior to the reaction the slide was washed with 50 mM Tris, 150 mM NaCl, 0.5 mM TCEP, pH 7.5. The UEV injection was performed with 2.5 µM UEV, 50 mM Tris, 150 mM NaCl, 0.5 mM TCEP, pH 7.5, 0.8% (w/v) d-glucose, 1 mM Trolox, 0.1 mg ml$^{-1}$ glucose oxidase (Sigma) and 0.02 mg ml$^{-1}$ glucose catalase (Sigma). The ubiquitination reaction injection was performed with 2.5 µM UEV, 2.2–4.4 µM RNF4 and 0–5.5 µM Ub~4xSUMO-2, 50 mM Tris, 150 mM NaCl, 0.5 mM TCEP, pH 7.5, 0.8% (w/v) d-glucose, 1 mM Trolox, 0.1 mg ml$^{-1}$ glucose oxidase (Sigma) and 0.02 mg ml$^{-1}$ glucose catalase (Sigma). An RNF4 construct containing full-length RNF4 linearly fused to a second RING domain was used in the ubiquitin reaction, similar to the substrate single turnover ubiquitination assay used to validate the proteins. One of the E2~Ub binding sites within the RING domain dimer was mutated (M140A and R181A) allowing only one E2~Ub conjugate to bind during the ubiquitination reaction. Ubiquitin within the Ubc13~Ub conjugate contains a K63R mutation such that only a single ubiquitin is transferred onto a substrate molecule, rather than forming ubiquitin chains. smFRET trajectories were acquired with 1 s integration time at 22 °C.

**Data processing**. IDL 6.0 was used to process an output containing intensity versus time for each single-molecule trajectory detected by the EMCCD camera. Data were analysed in Matlab as described previously[30]. FRET efficiency ($E_{FRET}$) was calculated from the raw Cy3B and AlexaFluor 647 intensity traces using:

$$E_{FRET} = I_D/(I_D + \alpha I_A) \qquad (1)$$

where $I_D$ = donor (Cy3B) intensity, $I_A$ = acceptor (AlexaFluor 647) intensity and $\alpha$ = 0.88 accounts for 12% leakage into the AlexaFluor 647 detection channel[30]. Histograms were prepared by averaging the first ten frames of each single-molecule trace separately and these averages were combined to produce the overall FRET population. Histograms were normalised to area unity to compare FRET populations between samples. Cross-correlation analysis of single-molecule trajectories of donor and acceptor dyes was carried out using the expression below implemented in matlab:

$$G(\tau) = \frac{\sum (I_D(t) - \overline{I_D})(I_A(t+\tau) - \overline{I_A})}{N \sum \overline{I_D} \overline{I_A}} \qquad (2)$$

where $I_D(t)$ and $I_A(t)$ are the donor and acceptor intensities at a given time point, $\overline{I_D}$ and $\overline{I_A}$ represent the mean donor and acceptor intensities over the entire single-molecule trajectory normalised to the total number of data points ($N$). The $G(\tau)$ function compares the intensity trace of the donor at a time $t$ and the acceptor trace at a time $t+\tau$.

**Reporting summary**. Further information on research design is available in the Nature Research Reporting Summary linked to this article.

## Data availability

Data underlying all Figures and Supplementary Figures are available in the source data file and at https://doi.org/10.17630/5f652bc1-1d66-400a-bd76-609b7de0bb25. All other data are available from the corresponding authors on reasonable request. Source data are provided with this paper.

## Code availability

Code used to generate 3D smFRET histograms in Fig. 2b, Supplementary Figs. 3 and 12a and contour plots in Fig. 5a, Supplementary Figs. 13a and 14 is included in the source data file and at https://doi.org/10.17630/5f652bc1-1d66-400a-bd76-609b7de0bb25. Source data are provided with this paper.

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

## Acknowledgements

We thank Anna Plechanovová and Gorjan Stojanovsky in the initial stage of the project. We thank Ulrich Zachariae of the University of Dundee for useful discussion. We thank the Division of Signal Transduction Therapy, University of Dundee for the gift of His-UBE1. This work was supported by an Investigator Award from the Wellcome Trust (098391/Z/12/Z) and (217196/Z/19/Z) and a Programme grant from Cancer Research UK (C434/A21747) to R.T.H.; J.C.P. thanks the University of St Andrews for financial support.

## Author contributions

E.B. cloned, expressed and purified proteins, conducted biochemical and smFRET experiments and interpreted data. J.C.P. conducted smFRET experiments and interpreted data. E.B. and J.C.P. contributed to data analysis. E.B., J.C.P. and R.T.H. wrote the paper. R.T.H. conceived the project and contributed to data analysis.

## Competing interests

The authors declare no competing interests.
