## [Peer Review File · Nature Communications]

Reviewers' comments:

Reviewer #1 (Remarks to the Author):

Branigan, Penedo and Hay describe a very interesting approach to monitoring the conformation of the ligated ubiquitin (Ub) attached to the Ub-conjugating enzyme (E2) using single molecule FRET (smFRET). They conclude that the Ub can have multiple conformations relative to the E2 prior to the presence of the ligase (E3), whereupon it adopts a closed conformation, and they propose that it is the closed state that leads to transfer of the Ub, triggered by binding of the RING domain in the E3 to the E2~Ub. The data is supportive, and the authors provide a clear description of the model they propose. The results strongly support the hypothesis that the closed conformation of Ub held close to the E2 results from the action of both RNF4 and UEV, and the 'closed state' is the effective conformation that leads to transfer to the Ub to substrate. As noted below, the only aspect that is insufficiently described is the dynamic aspect of interconversion of these states and how this corresponds to the reaction rate and the experimental detection rate of the states.

Comments/corrections:

The use of His-tagged Ub in the assays could potentially cause a problem with extracting the Zn bound in the RING. This happens over time (usually anywhere from ~1-4 hrs), such that this may not be a consequence in the authors reactions. However, this should be noted on the timescale of the incubation of Ub-His6 with the RING proteins.

Figure 1C. The visual description of WT -ATP and WT – UEV is difficult to easily recognize. It seems that the assays are done in the "absence" of ATP or UEV; however, the close spacing of the letters leaves this meaning obscure. The figure could be clarified.

Figure 1D. The figure panel does not make its point clearly. One presumes that the RING dimer is made up according to the WT:WT vs Mut:WT, vs Mut:Mut composition. However, it is not at all certain that this is correct, or how these compositional dimers were attained. The procedure is cited as ref 20. If the resolution is via fluorescence, then a minimal description is warranted. In an admixture of WT and Mut, how do they avoid contamination of the resultant 50% of WT:Mut with 25% Mut:Mut and 25% Wt:Wt?

Page 3, line 70-71: The presence of RNF4 RING increases the population of the high-FRET state, but it does not exist exclusively. How do the authors account for interconversion between states, such that the 'active' state may be one of the intermediately populated states that is then reacted out of this state, followed by repopulation in the distribution?

Page 5, line 111: It seems clear that the high FRET state is the 'closed' conformation; however, some comment on the timescale of the measurement is warranted to indicate that the deconvolution can be done as if the states are in slow exchange on this timescale. It is likely still a distribution of states exchanging at some rate. How do the measurements account for a case where a reaction occurs from the middle FRET state, followed by re-equilibration of the populations?

Page 7, line 158-160: The smFRET experiments are indeed quite powerful, however, they are not unperturbing and of rather coarse resolution. The absence of a dynamic aspect to conformational interchange remains. The impact of the dyes is reasonably accounted in the controls.

Reviewer #2 (Remarks to the Author):

How Ub is transferred from activated E2 to a substrate is a fundamental question in the field. In this work, Branigan et al., performed a series of elegant mechanistic studies, based on their previous structural analysis, to determine how RING E3 activates E2~Ub for transferring Ub to the substrate, and concluded that a 'closed' conformation of E2~Ub induced by binding to E3 is the active conformation for Ub transfer.

My major reservation about this work is, indeed their previous structural study suggests that the 'closed' conformation of E2~Ub is likely catalytically active and their kinetic results are mostly consistent, however the new data here do not actually provide a stronger support for this conclusion. The authors tried to support the relevance of the closed conformation in Ub transfer in primarily three ways: loss of ubiquitylation by I44A mutation on Ub, by L106A on Ubc13 and by adding RNF4. The hydrophobic patch on Ub is essential for most Ub transfer, and other mechanisms have been proposed. RNF4 and L106 on Ubc13 are essential for the closed conformation, however, their roles in promoting Ub transfer may be multiple and not necessarily due to the closed conformation per se. It would be helpful if the authors can demonstrate the sufficiency of this closed conformation in Ub transfer. For example, they showed that E2~Ub occupies the closed conformation at lower temperature even without RNF4. If the closed conformation is indeed sufficient, RNF4's contribution to Ub transfer should diminish at lower temperatures.

Other points:

1. Using FP to demonstrate ubiquitylation is acceptable but unconventional. Especially, in Fig. 1C and 2b, it is hard to understand why the curves of 'WT-UEV' and 'assorted mutants' quickly reach a plateau. Need to show the actual gel of the time course
2. It would be very helpful to have an illustration of the 'intermediate' conformation.
3. Fig. 3a, the Ex wavelength should be 532nm (?)
4. Fig. 3c, A647 seems to decay at the same rate before and after the injection, why?
5. Fig. 3. Loss cy3B does not necessarily mean that the Ub is transferred to the receptor. The control in which only SUMO-Ub is omitted is required.

Reviewer #3 (Remarks to the Author):

The authors test the hypothesis that Ub transfer from E2 to a substrate occurs through a closed conformation of E2 using single-molecule FRET. First, they constructed a fluorophore-labeled E2-Ub and measured FRET in dependence of E3 and UEV at different temperatures, and found a preference of the closed conformation for the fully assembled complex. When inactivating Ub by mutation, only the low FRET state was detected. Next, the authors followed ubiquitination in a real-time single-molecule FRET assay by attaching Ubc13-Ub to a glass surface and adding UEV and RNF4. Quantifying FRET loss after injection of the reaction partners, they showed that the reaction occurs out of the high-FRET (closed) conformation and determined reaction rates from dwell time analysis.

The study is well designed, the experiments are well described and important controls were made. Using both spatial and temporal information available from smFRET experiments, the authors were able to support the hypothesis of closed-state ubiquitin transfer.

Minor questions:

Were the FRET histograms in figure 2cd recorded at 22°C? (please add information to the caption, and to methods section)

In the real-time smFRET trajectories, I would expect to see different FRET states for Ubc13-Ub before and after addition of E3 and UEV – essentially sampling the conformational space in a similar way as shown in figure 1. Can the authors comment on this? I also do like the pie charts in figures 1 & 2, and I wonder whether the authors could add this information in figure 3.

Response to Reviewer's Comments:

Reviewer #1 (Remarks to the Author):

Branigan, Penedo and Hay describe a very interesting approach to monitoring the conformation of the ligated ubiquitin (Ub) attached to the Ub-conjugating enzyme (E2) using single molecule FRET (smFRET). They conclude that the Ub can have multiple conformations relative to the E2 prior to the presence of the ligase (E3), whereupon it adopts a closed conformation, and they propose that it is the closed state that leads to transfer of the Ub, triggered by binding of the RING domain in the E3 to the E2~Ub. The data is supportive, and the authors provide a clear description of the model they propose. The results strongly support the hypothesis that the closed conformation of Ub held close to the E2 results from the action of both RNF4 and UEV, and the 'closed state' is the effective conformation that leads to transfer to the Ub to substrate. As noted below, the only aspect that is insufficiently described is the dynamic aspect of interconversion of these states and how this corresponds to the reaction rate and the experimental detection rate of the states.

Comments/corrections:

The use of His-tagged Ub in the assays could potentially cause a problem with extracting the Zn bound in the RING. This happens over time (usually anywhere from ~1-4 hrs), such that this may not be a consequence in the authors reactions. However, this should be noted on the timescale of the incubation of Ub-His6 with the RING proteins.

We thank the reviewer for their comment regarding the stability of the Zn bound RING domain of RNF4 in the presence of his-tagged ubiquitin. Our single molecule FRET experiments of the stable isopeptide linked E2~Ub conjugate are carried out within about 30 minutes of the mixing his-tagged ubiquitin with the RING domain dimer of RNF4. Real-time single molecule FRET experiments using the reactive thioester linked conjugate were collected over a 5-minute interval during which time RNF4 is added to the conjugate. The extraction of Zn from the RING domain of RNF4 should not pose a problem in our single molecule FRET experiments. In addition the validation assay of the his-tagged and labelled E2~Ub conjugate compared to untagged and unlabeled shown in Supplementary Fig. 11, reports similar reactivity of tagged and untagged conjugates with RNF4 up to 10 mins incubation time. Fluorescence polarization experiments are performed using untagged ubiquitin. Taking into account the reviewers comment we have updated the methods section on page 30 to include the timescale of the incubation for the stable isopeptide linked E2~Ub conjugate in the single molecule FRET experiments.

Figure 1C. The visual description of WT -ATP and WT – UEV is difficult to easily recognize. It seems that the assays are done in the “absence” of ATP or UEV; however, the close spacing of the letters leaves this meaning obscure. The figure could be clarified.

The reviewer is correct in their interpretation of the graph of our fluorescence

polarization assay in Fig. 1c, however, we have modified the description of each experiment to clarify how the assays were performed. Descriptions now read as follows: “Reaction”, “Reaction - ATP” and “Reaction - UEV” (Fig 1). We have also included fluorescein imaging of SDS-PAGE analysis of these reactions in a new Fig. 1d, which contains the same naming strategy. In addition Coomassie stained SDS-PAGE analysis of these reactions is presented in a new Supplementary Fig. 1 on page 2 of the Supplementary Information.

Figure 1D. The figure panel does not make its point clearly. One presumes that the RING dimer is made up according to the WT:WT vs Mut:WT, vs Mut:Mut composition. However, it is not at all certain that this is correct, or how these compositional dimers were attained. The procedure is cited as ref 20. If the resolution is via fluorescence, then a minimal description is warranted. In an admixture of WT and Mut, how do they avoid contamination of the resultant 50% of WT:Mut with 25% Mut:Mut and 25% Wt:Wt?

We thank the reviewer for their comment and we have provided further information about the RING dimer in the figure legend (Fig. 1e) on page 16 and in the methods section on page 28. We avoid this mixing issue by using a genetically encoded linearly fused dimer of the RING domain of RNF4, which ensures that the RING domain is constitutively dimeric, as RING dimerization is required for binding the E2~Ub conjugate. The results for this experiment are now contained in Fig. 1e and we have also renamed the experiments as follows: RING^{WT}~RING^{WT}, RING^{Mut}~RING^{WT} or RING^{Mut}~RING^{Mut}. This clarifies that a WT and/or mutant form of the RING domain is used in the linearly fused dimer and ~ denotes linear fusion. This is consistent with the notation RNF4^{WT} and RNF4^{Mut}, which is already used in Figure 5a (previously Figure 4a) on page 23.

Page 3, line 70-71: The presence of RNF4 RING increases the population of the high-FRET state, but it does not exist exclusively. How do the authors account for interconversion between states, such that the ‘active’ state may be one of the intermediately populated states that is then reacted out of this state, followed by repopulation in the distribution?

The reviewers comment refers to the following statement:

“The addition of the RNF4 RING domain dimer to the Ubc13~Ub conjugate in complex with UEV at all temperatures, captures the high FRET state, consistent with stabilization of a closed or active conformation of the Ubc13~Ub conjugate”

To address the reviewers comment about interconversion between FRET states, we have added smFRET trajectories pertaining to the conjugate alone and in complex with UEV and the RNF4 RING dimer in Fig. 2c on page 17 and Supplementary Fig. 5a-c on page 8 of the Supplementary Information. Low-, mid and high-FRET complexes present in solution are static and do not often exchange within our measurement time window (~ 1 min) or in the case of real-time experiments, within the reaction timescale (~ 10-20 seconds at 4.4 μM RNF4). The remaining 1% of the molecules showed some dynamic exchange between FRET states, but even within

this minority, the interconversion events are very rare with one or two events before photobleaching occurred. To confirm that the observed long-lived low-, mid, and high-FRET levels indeed correspond to static conformations and there is no hidden interconversion dynamics, we carried out cross-correlation analysis of the donor and acceptor signals. The results are shown in Supplementary Fig. 6 on page 10 of the Supplementary Information and display a flat cross-correlation curve randomly distributed around the zero value. This confirms the absence of fast transitions between FRET states, at least within the time resolution of 100 ms we used for this measurements. The changes to the text describing the results of this additional analysis can be found on pages 5 and 6.

Page 5, line 111: It seems clear that the high FRET state is the ‘closed’ conformation; however, some comment on the timescale of the measurement is warranted to indicate that the deconvolution can be done as if the states are in slow exchange on this timescale. It is likely still a distribution of states exchanging at some rate. How do the measurements account for a case where a reaction occurs from the middle FRET state, followed by re-equilibration of the populations?

The reviewers comment refers to the following statement:

“Using the stable isopeptide linked Ubc13~Ub conjugate allowed us to identify the high FRET state as the closed conformation and indicated that both UEV and RNF4 were required to restrain the Ubc13~Ub conjugate in this conformation.”

This statement forms the introductory sentence to the description of the results real-time FRET analysis of single ubiquitination events mediated by a RING E3. Our interpretation of the reviewers comment is that they are asking for deconvolution of the rate of the real-time reaction, in order to determine whether a rate of change in FRET is effecting or incorporated into the rate of reaction. This is to account for a change in FRET from a low or intermediate FRET state to a high FRET state followed by reaction. We do not observe any clear examples of interconversion between FRET states before the reaction within our time resolution of 1s. In addition, the FRET labelled thioester linked E2~Ub conjugate is broken apart upon reaction and ubiquitin is transferred on to substrate which diffuses into solution and is no longer visualized. As a result, the FRET signal disappears completely after the reaction takes place and therefore we cannot observe re-equilibration of the FRET population. This is addressed on page 9.

To address the reviewer’s interest in the FRET population in the real-time reaction, we have generated contour plots containing all FRET containing molecules for the RNF4^{WT} and RNF4^{MUT} reaction, and the reaction in the absence of substrate. The results are shown in Supplementary Fig. 14 a, b and c on page 23 of the Supplementary Information along with representative smFRET trajectories for the RNF4^{MUT} reaction in Supplementary Fig. 15 a and b on page 24 of the Supplementary Information. We have also generated three histograms obtained from the overlapping molecules contained within the contour plots at 15 second intervals (0-15, 15-30 and 30-45 sec). The contour plots shows how the initial three states evolved over time. The histograms show how the high FRET state disappears, while the low and intermediate FRET states remain. The low FRET state seems to build up over time,

which is a result of molecules reacting and disappearing from the high FRET state, so the relative contribution of the low FRET state is higher (as in the contour plot). This analysis demonstrates the following important points:

1. The only population that disappears during the reaction is the high FRET state.
2. The low FRET state is an inactive state, the fact that it becomes a higher proportion of the total clearly indicates i) there is no direct reaction from this state (i.e. it is catalytically inactive) and ii) there is no transition from this state to the high-FRET state that takes place faster than our time-resolution, as this would result in a decrease in its contribution over time.
3. The intermediate FRET state contribution is almost constant with time.
4. As a result of the reaction from the high FRET state, the relative contribution of the low FRET state becomes higher over time. This is not observed for the RNF4^{MUT} or in the absence of substrate, where no reaction is taking place. This rules out the possibility that the disappearance of the high FRET molecules in the RNF4^{WT} reaction might be due to this state switching to the low FRET. Also, the lack of any observed interconversion dynamics in the wild type or mutant supports the idea that only ubiquitin transfer leads to an apparent accumulation of the low-FRET state.

We have incorporated these results into the Supplementary Information and the results are described in the main text on pages 10 and 11.

Page 7, line 158-160: The smFRET experiments are indeed quite powerful, however, they are not unperturbing and of rather coarse resolution. The absence of a dynamic aspect to conformational interchange remains. The impact of the dyes is reasonably accounted in the controls.

The reviewers comment refers to the following statement:

“In this study, smFRET provides access to the conformation of individual E2~Ub conjugates in real-time undergoing rapid substrate ubiquitination from a closed conformation in a RING E3 ligase dependent reaction”

This statement forms part of our Discussion. In our revision of the manuscript we have further discussed on page 12 the key part that previously published high resolution structural information has played in the interpretation of our smFRET results. We have also discussed the chemical biology tools that have been used previously to generate stable complexes for high resolution data acquisition. We have highlighted that in this real-time smFRET experiment have been able to use the native and reactive thioester linked E2~Ub conjugate which contains no modifications to the active site to access structural information about the ubiquitination reaction using a rapid real-time output. This experiment provides information about the RING E3 catalysed ubiquitination reaction that was not accessible using other biophysical methods.

We have previously addressed the dynamics of the stable isopeptide linked E2~Ub conjugate and the reactive thioester linked E2~Ub conjugate in the comments made

above.

Reviewer #2 (Remarks to the Author):

How Ub is transferred from activated E2 to a substrate is a fundamental question in the field. In this work, Branigan et al., performed a series of elegant mechanistic studies, based on their previous structural analysis, to determine how RING E3 activates E2~Ub for transferring Ub to the substrate, and concluded that a ‘closed’ conformation of E2~Ub induced by binding to E3 is the active conformation for Ub transfer.

My major reservation about this work is, indeed their previous structural study suggests that the ‘closed’ conformation of E2~Ub is likely catalytically active and their kinetic results are mostly consistent, however the new data here do not actually provide a stronger support for this conclusion. The authors tried to support the relevance of the closed conformation in Ub transfer in primarily three ways: loss of ubiquitylation by I44A mutation on Ub, by L106A on Ubc13 and by adding RNF4. The hydrophobic patch on Ub is essential for most Ub transfer, and other mechanisms have been proposed. RNF4 and L106 on Ubc13 are essential for the closed conformation, however, their roles in promoting Ub transfer may be multiple and not necessarily due to the closed conformation per se. It would be helpful if the authors can demonstrate the sufficiency of this closed conformation in Ub transfer. For example, they showed that E2~Ub occupies the closed conformation at lower temperature even without RNF4. If the closed conformation is indeed sufficient, RNF4’s contribution to Ub transfer should diminish at lower temperatures.

We previously reported a crystal structure of the Ubc13~Ub conjugate in complex with the RING domain dimer of RNF4 in the absence of UEV, where we observed a closed conformation of the conjugate (Branigan et al., 2015, NSMB and also shown in Figure 1b) even though ubiquitin transfer does not take place in the absence of the UEV in our biochemical assays (Fig. 1c and d). There is no evidence in our data presented here to suggest that the closed conformation is sufficient for ubiquitin transfer. Our data show that ubiquitin transfer with the E2, Ubc13, occurs from a closed conformation of the E2~Ub conjugate and requires the E3 ligase RNF4, the pseudo E2 (UEV) and a suitable substrate for nucleophilic attack of the thioester linkage. The substrate in this case is a mono ubiquitinated SUMO chain (Ub~4xSUMO-2). Our single molecule FRET analysis of the stable conjugate shown in Figure 2b (previously Fig. 1e) and Supplementary Fig. 3 confirm that the conjugate has the ability to access the closed conformation in the absence of RNF4, UEV and substrate. Our real-time reactions in Fig. 5a and Supplementary Fig. 13 also show that ubiquitin transfer to substrate occurs from the closed conformation in an RNF4, UEV and substrate dependent manner. Our data show that for a thioester linked E2~Ub conjugate, which has access to all three defined low, intermediate and high FRET states, the closed conformation (represented by the high FRET state) is not sufficient, although an obligatory step, for ubiquitin transfer. Our data show that all three proteins (RNF4, UEV and substrate) are required to promote ubiquitin transfer and that RNF4 binds and captures the closed conformation to orchestrate ubiquitin transfer to substrate with UEV to substrate from a closed conformation of the Ubc13~Ub conjugate. Taking into account the reviewer’s concern, we have clarified

in the text our reasoning for performing the smFRET experiments on pages 3 and 4 and our interpretation of the results in the discussion on page 12.

Other points:

1. Using FP to demonstrate ubiquitylation is acceptable but unconventional. Especially, in Fig. 1C and 2b, it is hard to understand why the curves of ‘WT-UEV’ and ‘assorted mutants’ quickly reach a plateau. Need to show the actual gel of the time course

We have previously published the fluorescence polarization (FP) assay as a method to monitor ubiquitin chain formation in a ubiquitination assay¹. However in response to the reviewer’s suggestion we have run SDS-PAGE of each of the FP assays under non-reducing and reducing conditions and we have imaged these gels using the fluorescein labeled ubiquitin followed by Coomassie staining and imaging. The SDS-PAGE analysis for the FP assay in Fig. 1c is shown in a new figure in Fig. 1d on page 15 and in Supplementary Fig. 1 on page 2 of the Supplementary Information. The SDS-PAGE analysis for the FP assay of mutants shown in Fig. 3b is presented in Supplementary Fig. 8a and b on page 14 of the Supplementary Information. SDS-PAGE analysis for the FP assay of mutants shown in Supplementary Fig. 7b is displayed in Supplementary Fig. 7c and d on page 12 of the Supplementary Information. The reaction without UEV reaches plateau quickly due to the formation of the fluorescein labeled thioester linked Ubc13~Ub conjugate, as ubiquitin is not transferred to K63 of the substrate in the absence of UEV (Fig. 1d). Here the thioester is visualized on non-reducing SDS-PAGE, while the thioester is destroyed under reducing SDS-PAGE analysis (Supplementary Fig. 1). The reactions containing mutant forms of Ub, Ubc13, UEV or RNF4 also reach plateau quickly due to the formation of the fluorescein labeled thioester linked Ubc13~Ub conjugate, as little or no ubiquitin is transferred to substrate (Supplementary Fig. 7c and d and Supplementary Fig. 8a and b). The results of this assay have been further described in the manuscript on page 3 and the experimental procedure is described in the methods section on page 27.

2. It would be very helpful to have an illustration of the ‘intermediate’ conformation.

We understand the reviewer’s desire to see an illustration of an intermediate conformation but this would be purely speculative, as we do not have a structure of this conformation. In an intermediate conformation ubiquitin is likely to occupy the space between ubiquitin in a closed conformation and an open conformation as shown in Fig. 2a (previously Figure 1d). However there are a large number of conformational possibilities between these two states and as the intermediate FRET state is occupied and only slightly destabilized even in the presence of UEV (Fig. 2b), this state is only partially represented by ubiquitin occupying a similar space as the UEV. The hydrophobic interaction between Ubc13 and ubiquitin appears to be involved in maintaining an intermediate conformation based on mutations, which reduce this interaction. However our data cannot determine if the intermediate FRET state observed for the E2~Ub conjugate alone and in complex with UEV and the RNF4 RING domain dimer corresponds to the same conformation of the E2~Ub conjugate. Therefore we do not want to limit the potential space which ubiquitin can

occupy with respect to Ubc13 in an intermediate conformation using an illustration, as multiple conformations may actually make up this population. Instead we have provided further description in the manuscript about the potential for an intermediate conformation on pages 4 and 6.

3. Fig. 3a, the Ex wavelength should be 532nm (?)

The reviewer is correct. We thank them for their correction and we have made the appropriate changes to Fig. 4a (previously Fig. 3a) on page 21.

4. Fig. 3c, A647 seems to decay at the same rate before and after the injection, why?

We thank the reviewer for pointing this out. We have checked the injection start time and found it was indicated incorrectly in the Figure. We have now corrected the positioning of the injection time-window, which can be seen in Fig. 4c (previously Fig. 3c) on page 21. The injection start time now coincides with a faster rate of loss of AlexaFluor 647.

5. Fig. 3. Loss cy3B does not necessarily mean that the Ub is transferred to the receptor. The control in which only SUMO-Ub is omitted is required.

We agree with the reviewer that this is an important control to demonstrate that ubiquitin is only transferred to the substrate (Ub~4xSUMO-2). We have performed this experiment and show that no reaction takes place in the absence of substrate, which confirms that ubiquitin transfer is specific for the substrate and ubiquitin is not transferred to any other components of the reaction mixture. The results of this reaction are shown in Supplementary Fig. 13 on page 21 of the Supplementary Information. The contour plot of FRET trajectories (Supplementary Fig. 13a) and cumulative intensity plots of fluorescent spots over time (Supplementary Fig. 13b) show no loss of FRET and fluorescence intensity during the real-time reaction. smFRET dwell-time analysis (Supplementary Fig. 13c) shows similar rates of background Cy3B and AlexaFluor 647 photobleaching events to those seen in other control experiments presented in Fig. 4 and Fig. 5. The results of this experiment are described in the manuscript on page 10.

Reviewer #3 (Remarks to the Author):

The authors test the hypothesis that Ub transfer from E2 to a substrate occurs through a closed conformation of E2 using single-molecule FRET. First, they constructed a fluorophore-labeled E2-Ub and measured FRET in dependence of E3 and UEV at different temperatures, and found a preference of the closed conformation for the fully assembled complex. When inactivating Ub by mutation, only the low FRET state was detected. Next, the authors followed ubiquitination in a real-time single-molecule FRET assay by attaching Ubc13-Ub to a glass surface and adding UEV and RNF4. Quantifying FRET loss after injection of the reaction partners, they showed that the reaction occurs out of the high-FRET (closed) conformation and determined reaction

rates from dwell time analysis.

The study is well designed, the experiments are well described and important controls were made. Using both spatial and temporal information available from smFRET experiments, the authors were able to support the hypothesis of closed-state ubiquitin transfer.

Minor questions:

Were the FRET histograms in figure 2cd recorded at 22°C? (please add information to the caption, and to methods section)

The smFRET experiments in Fig. 2 c and d (now Fig. 3 c and d) were performed at 22 °C and we have updated the figure caption on page 18 and the experimental procedure in the methods section on page 30 to include the temperatures at which these experiments were performed. In light of the reviewer's request we have also updated the experimental procedure for the real-time smFRET experiments to include the temperature at which they were performed, which was also 22 °C. This can be found in the methods section on page 31.

In the real-time smFRET trajectories, I would expect to see different FRET states for Ubc13-Ub before and after addition of E3 and UEV – essentially sampling the conformational space in a similar way as shown in figure 1. Can the authors comment on this? I also do like the pie charts in figures 1 & 2, and I wonder whether the authors could add this information in figure 3.

We have directly addressed this comment by adding a histogram of the FRET population of the thioester linked E2~Ub conjugate along with the corresponding pie chart in Supplementary Fig. 12 on page 20 of the Supplementary Information. We have also prepared a 3D plot to demonstrate the difference between the FRET populations of the stable isopeptide linked E2~Ub conjugate and the reactive thioester linked E2~Ub conjugate, where the high FRET state is more populated for the thioester linked conjugate than the isopeptide linked conjugate. This may be a consequence of differences in the length and precise geometry of the linkage between ubiquitin and Ubc13 in the isopeptide mimic and the native thioester bonds. We have described these results on page 8 of the manuscript.

With respect the different FRET states upon addition of RNF4 and UEV, we have generated contour plots of all FRET containing molecules in Supplementary Fig. 14. We also generated three histograms and corresponding pie charts obtained from the overlapping molecules contained within the contour plots at 15 second intervals (0-15, 15-30 and 30-45 sec). The contour plots shows how the initial three states evolved over time. The histograms show how the high FRET state disappears, while the low and intermediate FRET states remain. The low FRET state seems to build up over time, which is a result of molecules reacting and disappearing from the high FRET state, so the relative contribution of the low FRET state is higher (as in the contour plot). We have incorporated these results into the Supplementary Information and the results are described in the main text on pages 10 and 11.

In response to a similar query from Reviewer 1 we have added representative traces for the isopeptide linked E2~Ub conjugate to demonstrate stability of the defined low, intermediate and high FRET states in the smFRET trajectories. Please find the response to Reviewer 1 on the bottom of page 2 and continued into page 3.

- 1 Branigan, E., Plechanovová, A. & Hay, R. T. Methods to analyze STUbL activity. *Methods in Enzymology* **618**, 257-280, doi:10.1016/bs.mie.2018.11.005 (2019).

REVIEWERS' COMMENTS:

Reviewer #1 (Remarks to the Author):

All responses are reasonable, and the manuscript should be published as is.

The authors have made clear and exceptional responses to all of the questions raised. In so doing, the manuscript is strengthened, the concepts made clearer, and the value and impact of the manuscript are advanced.

R. Andrew Byrd

Reviewer #2 (Remarks to the Author):

I thank the authors for the revision. My concerns have been addressed, errors corrected. I am now OK with publication